# Treatment Options for Elderly/Unfit Patients with Chronic Lymphocytic Leukemia in the Era of Targeted Drugs: A Comprehensive Review

**DOI:** 10.3390/jcm10215104

**Published:** 2021-10-30

**Authors:** Alberto Fresa, Francesco Autore, Eugenio Galli, Annamaria Tomasso, Luca Stirparo, Idanna Innocenti, Luca Laurenti

**Affiliations:** 1Sezione di Ematologia, Dipartimento di Scienze Radiologiche ed Ematologiche, Università Cattolica del Sacro Cuore, 00168 Rome, Italy; alberto.fresa92@gmail.com (A.F.); annamariatomasso2@gmail.com (A.T.); luca.stirparo@gmail.com (L.S.); 2Dipartimento di Diagnostica per Immagini, Radioterapia Oncologica ed Ematologia, Fondazione Policlinico Universitario A. Gemelli IRCCS, 00168 Rome, Italy; francesco_autore@yahoo.it (F.A.); eu.galli@live.it (E.G.); idanna.innocenti@yahoo.it (I.I.)

**Keywords:** chronic lymphocytic leukemia, elderly, comorbidities, targeted therapy

## Abstract

Chronic lymphocytic leukemia (CLL) incidence increases with age reaching 37.9/100,000 in patients over 85 years. Although there is no standardized geriatric tool specifically validated for CLL, a correct framing of the fitness status is of critical importance to individualize treatment strategies. Based on the evidence available to date, frontline chemoimmunotherapy has an increasingly narrowing application, being eligible for candidacy only in elderly fit patients without or with minimal geriatric syndromes. On the other hand, treatment with BCR inhibitors, monotherapy, or in combination with anti-CD20 antibodies (e.g., obinutuzumab), must be preferred both for frontline and relapsed CLL not only in unfit patients, but also in fit patients with unmutated IGHV or harboring del(17p) and/or TP53 mutations/deletions. Second-generation inhibitors (e.g., acalabrutinib, zanubrutinib, pirtobrutinib) are novel compounds that, due to their better safety profile and different specificity, will help physicians overcome some of the safety issues and treatment resistances. In the era of targeted therapies, treatment decisions in elderly and/or unfit patients with CLL must be a balance between efficacy and safety, carefully evaluating comorbidities and geriatric syndromes to ensure the best approach to improve both quality of life and life expectancy.

## 1. Introduction

The incidence of chronic lymphocytic leukemia (CLL) is 3.5 per 100,000 people/year and significantly increases with age. The median age at diagnosis is 70 years, and significantly, men develop CLL more frequently in comparison with women (1.9-fold higher incidence) [1]. In patients above 65 years of age, incidence increases to 27.5 per 100,000 people and year, arriving at 37.9 per 100,000 people/year in the population over 85 years. Increased life expectancy results in an inevitably aging population; therefore, during recent years, specific studies have focused on the management of older and unfit patients. Currently, in CLL treatment guidelines [2], patients are classified as elderly if they are older than 65 years of age. Assessment of fitness status is difficult to standardize because different rating scales may stratify patients differently. When considering patients with comorbidities in clinical trials, the fitness is assessed on the basis of Cumulative Illness Rating Scale (CIRS) [3] and creatinine clearance (CrCl), calculated according to Crockoft–Gault [4]: patients are defined as fit if CIRS is less than or equal to 6 and CrCl is greater than or equal to 70 mL/min; therefore, patients with CIRS > 6 and/or CrCl < 70 mL/min are defined as unfit [5,6]. In recent years, the development of new non-chemotherapeutic drugs with safer toxicity profiles and proven efficacy has led to a shift in the treatment approach to the elderly and/or unfit population with CLL. Chemoimmunotherapy is currently offered to fit patients, using less-toxic treatment regimens in elderly patients (e.g., fludarabine–cyclophosphamide–rituximab for young patients, bendamustine–rituximab for elderly patients) [2]. The only exception is for unfit patients treated with the chlorambucil–obinutuzumab combination scheme, which showed greater efficacy in the CLL11 trial [5] (see Section 3.1 for further details). Target therapies (i.e., ibrutinib, acalabrutinib, idelalisib, venetoclax) are indicated in unfit patients and in those with adverse prognostic factors, i.e., deletion of the chromosome 17 [del(17p)] and/or TP53 gene mutations, immunoglobulin heavy chain variable (IGHV) gene mutation status unmutated (see Section 3.2 and Section 3.3 for further details) [2]. The goal of treatment in elderly/unfit patients is effective control of the disease, while in younger patients, the disease eradication, defined as the achievement of minimal residual disease (MRD) negativity, is a priority. Undetectable MRD (uMRD), or MRD negativity, is defined as the presence of less than 1 CLL cell per 10,000 leukocytes (<10–4) and can be determined on peripheral or bone marrow blood; in recent years, clinical trials have focused much attention on uMRD because its achievement has shown a positive predictive value by improving the long-term outcome of patients in terms of progression-free survival (PFS, length of time from the treatment start to disease progression or death) and overall survival (OS, length of time from the treatment start to death) [7]. Although a significant increase in progression-free survival has been observed with both fixed-duration and continuous therapies compared with the past, MRD negativity is primarily achieved with time-limited therapies; continuous therapies, on the other hand, are unlikely to achieve uMRD, but considering the toxicity profile and type of administration, may be more suitable for specific subgroups of patients. In this context, we will analyze the impact of the different types of therapies on elderly and/or unfit patients, considering their characteristics and evaluating which may be in real life the best approach to improve both the quality of life and the life expectancy of these patients.

## 2. The Evaluation of Patients’ Fitness Status

A higher burden of comorbidities is known to impact the outcome of patients with CLL [8,9,10], and specific recommendations for elderly patients should be considered in clinical practice [11]. Assessment of fitness status in hematologic patients can make use of multiple rating scales, some of which are composed of multiple scores. As mentioned above, the definition of “patient with comorbidity” in clinical trials is that of a patient with CIRS > 6 and/or CrCl < 70 mL/min. This definition provides quantitative ratings of chronic illness burden, mainly considering the impairment of different organs. In fact, the CIRS assigns a level of severity to each system from 0 (no problem) to 4 (severe impairment) depending on the impairment, for while creatinine clearance provides a more precise assessment of renal impairment. In addition, in clinical practice, it is of fundamental importance to assess the performance status of the patient. Although both the Eastern Cooperative Oncology Group Performance Status (ECOG PS) [12] and the Karnofsky Performance Status [13] are valid scores for this assessment, historically, patients with CLL have been evaluated with the ECOG PS, since it provides not only information on ability to care for themself and daily activity, but also physical ability, such as walking or working. In particular, ranging from 0 (fully active) to 5 (dead), patients with an ECOG PS greater than or equal to 2 are considered patients who require assistance that would define them as not fully self-sufficient. More sophisticated assessments resulting from the combination of several scores are the Comprehensive Geriatric Assessment (CGA) [14] and the Geriatric Assessment in Hematology (GAH scale) [15], whose composition and impact on clinical activity will be discussed later. Finally, the presence and compliance of caregivers should be taken into account for this category of patients. Although it is not commonly taken into account in clinical trials and, therefore, there are no validated data, the presence of a reliable caregiver for elderly/unfit patients is important for the candidacy to different treatments and the effective adherence of patients to treatment.

### 2.1. Age, CIRS, ECOG, Creatinine Clearance

Several retrospective analyses have investigated the impact of fitness status in terms of CIRS, renal impairment (CrCl < 70 mL/min), and ECOG PS in CLL patients that have undergone chemoimmunotherapy (CIT) [16,17,18,19,20,21]. For elderly/unfit CLL patients, direct and indirect comparisons have been made, mostly with the following regimens: fludarabine–cyclophosphamide +/− rituximab, bendamustine–rituximab, chlorambucil–rituximab, and ibrutinib. What emerged is that for patients that had undergone CIT, in most of them, neither older age nor higher ECOG PS had an impact on survival, while higher CIRS significantly worsened prognosis in almost all of them [16,17,18,19]. In only two studies, both investigating the use of bendamustine–rituximab in real life, a high ECOG PS showed a negative impact on survival in the Swedish group study in multivariate analysis (141 patients, of which 20% ECOG ≥ 2) [20], while in the GIMEMA-ERIC and US group study, only in univariate analysis (165 patients, of which 10% ECOG ≥ 2) [21]. For ibrutinib, Gordon et al. showed that older age and a higher burden of comorbidities (expressed both in terms of CIRS ≥ 7 and CIRS3+) appear to have a negative impact on survival, whereas ECOG does not. In their cohort of 145 patients, the median age was 70 years and 92% of patients had an ECOG < 2 [10]. In a more recent retrospective multicentre study on 712 CLL patients treated with ibrutinib outside clinical trials (median age 70 years with 68% of patients ≥65 years old, and 15.6% of patients with an ECOG-PS ≥ 2), Tedeschi A. et al. confirmed and enriched these data. They showed that older age does not independently influence survival and ibrutinib tolerance and confirmed that CIRS > 6 is a predictor of a poorer progression-free survival (PFS) and event-free survival (EFS). Moreover, they showed that the only baseline parameters affecting overall survival are ECOG-PS ≥ 2 and neutropenia [22]. An important consideration to make is that studies in which ECOG showed a negative impact included more unfit and, on average, older patients than those in which a higher CIRS showed an impact. Arguably, the burden of comorbidity impacts more when comparing fit and unfit patients, even if elderly, whereas significance flattens when comorbidities are homogeneously distributed across the entire population. However, collectively, these studies suggest that with a better profiling of fitness status at the moment of treatment, clinicians can lead CLL patients to more effective therapeutic choices.

### 2.2. Comprehensive Geriatric Assessment

The multidimensional Comprehensive Geriatric Assessment (CGA) has been shown to predict outcome in elderly cancer patients [14]. Although there is lack of consensus on what parameters to evaluate and their predictivity, assessing geriatric syndromes in the oncohematologic population prior to the initiation of treatment is of paramount importance to best predict mortality, treatment-related toxicity, treatment completion/compliance, and healthcare utilization. The main components of comprehensive geriatric assessment (CGA) that should be evaluated are functional capacity, fall risk, cognition, mood, polypharmacy, social support, financial concerns, goals of care, and advance care preferences. A recent systematic review of 44 studies showed that the most common geriatric impairments found through studies performed on various oncohematologic populations are polypharmacy (in a median of 51% of patients), risk of malnutrition (median 44%), instrumental activities of daily living (IADL) impairments (median 37%), impaired physical capacity (median 27%), activity of daily living (ADL) impairments (median 18%), symptoms of depression (median 25%), and cognitive impairment (median 17%). Frailty, as well as IADL functioning, nutritional status, and polypharmacy, have been shown to impact mortality, toxicity, and treatment discontinuation [23]. In this context, to the best of our knowledge, the Geriatric Assessment in Hematology (GAH scale) is the only validated tool developed in a prospective multicenter observational study specific for patients with hematological malignancies [15]. It is composed of 30 items belonging to 7 dimensions: polypharmacy, gait speed, mood, ADL, subjective health status, nutrition, mental status, and comorbidities. Although it has been shown to have a predictive power overlapping with that of CGA in cancer patients, the timing of administration (10–12 min) could lead to limit its use in daily clinical practice. A score of more rapid use, developed by combining ADL, IADL, and CIRS, and evaluated on a population of patients with CLL but not yet validated in a clinical trial, showed a prognostic value correlating a worse performance with shorter survival [24].

## 3. Therapeutic Approaches in Older and/or Unfit Patient Populations

### 3.1. Chemoimmunotherapy

Currently, as the options for first-line chemo-free treatments are increasingly expanding, the role of chemoimmunotherapy, especially in the geriatric population, is decreasing. The most relevant trials that have evaluated chemoimmunotherapy regimens with a focus on the elderly/unfit population are shown in Table 1. Fludarabine-based regimens, previously used even in the elderly population if considered fit (CIRS < 6 and CrCl > 70 mL/min), at full or even reduced doses, now have a very limited role [25]. For patients who are 65 years of age or older, there is still the option of chemoimmunotherapy treatment with bendamustine in combination with an anti-CD20 antibody, mostly rituximab, but only if considered fit. The phase 3 trial CLL10 [25] in fit patients with untreated CLL and little comorbidity burden (median age 61 years, total CIRS score 2) demonstrated that bendamustine–rituximab (BR), despite being inferior to full dose fludarabine–cyclophosphamide–rituximab (FCR) in terms of PFS, has a favorable risk–benefit profile in patient aged 65 or older thanks to its better toxicity profile (median PFS 49 months, grade ≥ 3 infections 27%) [25]. Compared with chlorambucil plus rituximab (R-Chl) in patients unsuitable for FCR (MABLE trial) [26], BR showed a better median PFS (40 versus 30 months), but worse toxicity profile (grade ≥ 3 infections 19% versus 10%) [26]. To date, no prospective trials supporting reduced relative dose intensity (RDI, the ratio of the dose actually delivered over time to the standard dose) of BR schedules (e.g., bendamustine dosed with 70 mg/m^2^, reduced number of total cycles in patients achieving a complete response) have been performed. However, retrospective multicenter experiences in elderly and/or unfit CLL patients in Italy before the approval of chlorambucil–obinutuzumab (G-Chl) regimen have highlighted and evaluated the need for dose adjustment of the previously used chemoimmunotherapy regimens [27,28,29]. They showed that patients obtaining a complete response to treatment could benefit from lighter regimens (e.g., by reducing the dose per cycle or the number of cycles), but otherwise, RDI reduction significantly affects both PFS and OS [9]. In addition, the combination of bendamustine and obinutuzumab in patients with untreated CLL has been investigated in a nonrandomized trial, GREEN [30] (median age 68 years). This treatment showed a complete remission rate of 32%, but also the occurrence of toxicities as tumor lysis syndrome [30]. The efficacy of G-Chl regimen in CLL patients with comorbidities has been assessed in CLL11 trial [5,31], which led to its approval by FDA and EMA as a first-line treatment in patients with comorbidities (CIRS > 6 and/or CrCl < 70 mL/min) and in real-life studies [32,33]. The CLL11 study also showed that Chl-R achieves longer PFS (median 16 versus 11 months) and overall survival (OS) than chlorambucil monotherapy. Addition of the newer CD20 antibody obinutuzumab to chlorambucil resulted in an improvement of both PFS and time-to-next-treatment (medians: 51 versus 38 months). Rates of grade 3–5 infections were not significantly different between the three study arms (10–15%), but infusion-related reactions (IRR) occurred more frequently with G-Chl than with R-Chl (grade 3–4 IRR 20% versus 4%) [5]. The addition of ofatumumab to bendamustine [34] or chlorambucil [35] showed convincing results in the elderly/unfit population. Nevertheless, even if no direct comparison of obinutuzumab with ofatumumab has been performed within a controlled randomized trial, in a Bayesian network meta-analysis chlorambucil plus obinutuzumab showed superior efficacy in comparison with chlorambucil plus ofatumumab or rituximab [36]. Chlorambucil monotherapy and/or in combination with rituximab (or steroid therapy) maintains a value mostly as a historical reference to compare new treatments in this specific subgroup of patients; although, in some centers, it is still used albeit minimally due to necessity, heavy impairment of fitness status or logistical difficulties of the patient. Obviously, for patients with TP53 mutations/deletions, the chemoimmunotherapy option should not be considered due to the advent of new biological drugs that have shown superiority in terms of outcome in this subgroup of patients.

### 3.2. B-cell Receptor (BCR) Inhibitors

#### 3.2.1. Ibrutinib

Ibrutinib is an inhibitor of Bruton’s tyrosine kinase (BTKi), a nonreceptor tyrosine kinase critical for BCR signal transduction, which has been approved for the treatment of relapsed/refractory CLL, and also in those harboring del (17)/TP53 mutations in the first-line setting. Data from trials that evaluated the impact of immunotherapy on the elderly/unfit population are shown in Table 2. In older patients (median age 71 years), it has been explored in a small nonrandomized trial as frontline therapy and has demonstrated activity and a favorable toxicity profile [37]. Concerning the impact of comorbidities, in the two main clinical trials where ibrutinib demonstrated its superiority in CLL patients against chlorambucil frontline (RESONATE-2) [38] and ofatumumab in relapsed/refractory patients (RESONATE) [39,40], 32% and 20% of patients with CIRS > 6 were enrolled, respectively. Although demonstrating the superiority in terms of overall response rate (ORR), progression-free survival (PFS), and overall survival (OS), specific side effects of ibrutinib have been identified. The side effects that represent a main concern in the elderly/unfit population are mainly atrial fibrillation, hypertension, and bleedings. Considering the low prevalence of older and/or comorbid patients in clinical trials, the real-life experiences with ibrutinib in this population have to be considered. A retrospective analysis of the French FILO group showed that ibrutinib in very elderly patients (age above 75 years old) is feasible and effective in this group of patients [41]. In particular, they showed response rates superimposable to that of clinical trials, PFS 77% at 12 months, an estimated median OS of 21 months, and mostly low-grade and manageable adverse events, especially within the first 6 months: bleeding (19%), cardiac toxicity (7%), diarrhea (24%), and myalgia/arthralgia (20%). While age per se does not have an impact on the outcome of patients treated with ibrutinib, comorbidities do affect OS, PFS, and EFS. In particular, in a retrospective multicentre study [10], Gordon et al. showed the impact of CIRS ≥ 7 and the severe impairment of a single organ (CIRS3+) affecting OS (2-year OS 79% versus 100%) and EFS (median 24 versus 37 months), both in frontline and relapsed CLLs [10]. Tedeschi et al., in their retrospective multicentre study [22] on 712 patients on treatment with Ibrutinib, showed that ECOG-PS ≥ 2 and neutropenia are the only baseline parameters affecting overall survival. Moreover, since it is metabolized via the CYP3A4 enzyme system, they demonstrated that the influence of concomitant medications and CYP3A-inhibitors (e.g., clarithromycin, erythromycin, diltiazem, itraconazole, ketoconazole, ritonavir, verapamil) with ibrutinib has no impact on outcome, though CYP3A4 inhibitors use severe comorbidities correlated with an increased risk of permanent dose reduction. Besides, concerning the discontinuation of treatment, definitive discontinuation of treatment due to toxicity negatively affects OS independently from age, while permanent dose reduction did not have any OS impact [22].

#### 3.2.2. Acalabrutinib

Acalabrutinib is a second generation, irreversible BTKi approved for the treatment of CLL with greater selectivity for BTK than ibrutinib. The efficacy of acalabrutinib has been investigated in phase III trials both frontline (ELEVATE-TN) [42] and in relapsed/refractory (ASCEND) [43] CLL. In the ELEVATE-TN trial [42] acalabrutinib +/− obinutuzumab showed better results than G-Chl in elderly/unfit patients (median age 70 years, 84% aged 65 years or older) in terms of ORR and PFS, but not OS (median follow-up 28.3 months). In the ASCEND trial [43] (median age 67 years; >75 years old 22%), acalabrutinib monotherapy showed a better median PFS (not reached versus 16.5 months) than BR and idelalisib–rituximab. Moreover, no difference in terms of PFS was observed between mutated and unmutated IgVH. Serious adverse events occurred in 29% of patients treated with acalabrutinib monotherapy, 56% with ibrutinib–rituximab, and 26% with BR. To note, while in the aforementioned trials atrial fibrillation rates with ibrutinib were 16% and 11%, respectively, acalabrutinib showed much lower incidence rates (1% and 5%, respectively). Treatment discontinuations due to toxicity in the ELEVATE-TN and ASCEND were 9% and 11% at the median follow-up of 28.3 and 16.1 months, respectively. Acalabrutinib has also demonstrated efficacy and tolerability in ibrutinib-intolerant patients with CLL [44]. Recently, a noninferiority direct comparison of the relative benefit-to-risk profiles of acalabrutinib versus ibrutinib was made in a randomized phase 3 trial [45] to test the hypothesis that acalabrutinib was noninferior to ibrutinib in PFS with improved tolerability. Median age was 66 years, with patients >75 years old at 16.4%. After a median follow-up of 40.9 months, patients treated with acalabrutinib showed a superimposable outcome, with a median PFS of 38.4 months in both arms. The incidence of atrial fibrillation/flutter was significantly lower with acalabrutinib versus ibrutinib (9.4% vs. 16.0%); incidence of infections, Richter transformation, and deaths were comparable in both arms. Concerning bleeding events, even if the occurrence was collectively less frequent with acalabrutinib (38%) versus ibrutinib (51.3%), major bleedings were comparable (4.5% versus 5.3%, respectively) [45]. To date, no real-life data are available on the use of acalabrutinib in CLL patients.

#### 3.2.3. Idelalisib–Rituximab

Idelalisib, a phosphoinositide 3-kinase (PI3K) inhibitor whose efficacy has been demonstrated in combination with rituximab, is another valid treatment option in these patients. A nonrandomized phase 2 study [46] conducted in elderly CLL patients (median age 71 years) showed that frontline treatment with idelalisib–rituximab (Idela-R) is feasible in these patients, with ORR of 97%, including 19% complete responses, and a three-year OS of 83%. Nevertheless, it has also shown a rather high toxicity profile, with the most frequent and invalidating amongst adverse events being diarrhea and/or colitis (64%), pyrexia (42%), and pneumonia (19%). In a phase 3 trial [47,48], compared with rituximab monotherapy, Idela-R demonstrated a superior efficacy in relapsed/refractory CLL (median age 71 years, total CIRS score 8), with better PFS (medians not reached versus 6 months) and OS (12-months OS 92% versus 80%). However, the use of Idelalisib has declined over the years due to toxicities associated with its use, mainly autoimmune diarrhea and colitis, pneumonia, and hepatitis. These side effects, which typically occur later in life, can be disabling and often lead to discontinuation of treatment, especially in elderly/unfit patients. However, it should be considered that drug-related autoimmune disorders appear to occur less frequently in elderly or pretreated patients (<1%), with no impact on survival [49].

### 3.3. Venetoclax

Venetoclax is an inhibitor of the antiapoptotic Bcl-2 protein. The first efficacy data, which led to its approval in CLL patients, were derived from a phase II trial (median age 67 years) evaluating venetoclax monotherapy in relapsed CLL with del(17p) [50]. ORR was 79% and 1-year PFS was 72%. Since then, several trials have investigated the combination of venetoclax and CD20-antibody. The phase III trial MURANO [51] showed better outcomes with the combination of venetoclax–rituximab compared with BR, with sustained PFS and OS benefits (HR for PFS 0.17, CI 0.11–0.25; *p* < 0.0001). Moreover, the recent 5-year update showed that the higher rate of undetectable MRD (uMRD) (70%), 38% of which remained uMRD at 5 years, was associated with improved OS [7]. Then, recently, another phase 3 study showed the best outcome in terms of PFS (2-year PFS 88.2% versus 64.1%) of frontline fixed-duration venetoclax–obinutuzumab treatment compared with chlorambucil-obinutuzumab in patients with CLL with coexisting conditions (CIRS > 6 and/or CrCl < 70 mL/min, median age 72 years) [6]. These seminal studies led to FDA and EMA approval of the following indications for the use of venetoclax in CLL: monotherapy in the presence of 17p deletion or TP53 mutation in patients who are unsuitable for or have failed a BCR inhibitor, or in the absence of 17p deletion or TP53 mutation in patients who have failed both chemoimmunotherapy and a BCR inhibitor; in combination with rituximab in patients who have received at least one prior therapy; in combination with obinutuzumab in previously untreated patients. In the real-life setting, large retrospective multicentre studies, conducted in the United States of America [52,53], the UK [54,55], and Italy [56] have demonstrated reassuringly similar efficacy and survival to trial outcomes. A safety analysis was performed in a retrospective cohort study including 297 all-age CLL venetoclax-treated patients outside of clinical trials [53]. They confirmed the low-toxicity profile of venetoclax, with an incidence of clinical and laboratory tumor lysis syndrome (TLS) of 2.7% and 5.7% (independently predicted by TLS risk group and CrCl), and the following incidence of the most common grade 3/4 adverse events: neutropenia (39.6%), thrombocytopenia (29.2%), infection (25%), neutropenic fever (7.9%), and diarrhea (6.9%). In this cohort, 7.4% of patients discontinued venetoclax due to an adverse event. To evaluate efficacy and safety in older patients, a large retrospective real-life study on 342 relapsed/refractory CLL patients was conducted, comparing patients aged 75 years or older with patients younger than 75 years [55]. No differences in terms of overall response rates (81% versus 82%), 1-year PFS (79% versus 73%), or 1-year OS (77% versus 83%) were observed, not even by performing an analysis adjusted for the combination of venetoclax with antiCD20 antibodies. Toxicity was superimposable between the two groups, with TLS being 3% in both arms; further, although higher rates of grade ≥ 3 thrombocytopenia and neutropenia were observed in older patients, this did not clearly translate into higher rates of neutropenic infection (9% <75 versus 4% ≥75 years). The only difference with a tendency to statistical significance was reported for discontinuation due to toxicity, with a rate slightly higher for the ≥75 years group (6.6% versus 14%, *p* = 0.07) [55]. Concerning the impact of the fitness status, a retrospective multicentre Italian study investigated whether ECOG-PS and comorbidities retained a predictive value in a cohort of 158 patients (median age 70 years) treated with venetoclax outside clinical trial. It emerged that ECOG ≥ 2 was the only significant factor related to fitness, independently leading to lower PFS, EFS, and OS, while CIRS, CIRS3+, and concomitant medications did not affect patients’ management and outcomes [56].

### 3.4. Future Perspectives

Among all the new molecules under study at the moment, in our opinion, there are two drugs that more than any other are poised to become part of the future treatment algorithm for elderly and/or unfit CLL patients—the BTK inhibitors zanubrutinib and pirtobrutinib (formerly known as LOXO-305). Zanubrutinib is a highly specific, covalent BTKi, with a decreased inhibition of off-target kinases. To date, four clinical trials have shown the safety and efficacy of zanubrutinib in CLL. The initial Phase 1 trial showed brilliant responses in 94 patients with naïve or relapsed/refractory CLL/SLL (ORR 96.2%), without differences based on high-risk genetic features (ORR 100% in del(17p) and TP53 mutation) [57]. A phase 2 study conducted in China enrolled 91 relapsed/refractory patients with a median age of 61 years (range, 35–87). The ORR was superimposable across all subgroups analyzed, including patients harboring del(17p) and/or TP53 mutation (24%), which achieved high response rates of 91% (95% CI, 70.8% > 98.9%). PFS at 24 months was 80.5%, while at 36 months, it was 68.1% [58]. The Phase 3 Sequoia trial (NCT03336333) is currently comparing zanubrutinib with BR in patients with treatment-naïve CLL. The first results have been reported only for the nonrandomized Cohort C, enrolling 109 patients with del(17p) treated with zanubrutinib. The ORR was 94.5%, and the estimated 18-month PFS rate was 88.6% [59]. Follow-ups for Cohorts A and B have not been reported to date, as well as the results of Cohort D, which is studying zanubrutinib in combination with venetoclax. Finally, the interim analysis with the results of the ALPINE study, a phase 3 randomized trial comparing zanubrutinib and ibrutinib in patients with relapsed/refractory CLL, was recently presented, showing its superiority in terms of efficacy and safety in these patients. Particularly, in a population composed by 62.3% versus 61.5% patients of age ≥65 years for zanubrutinib and ibrutinib, respectively, ORR was significantly higher with zanubrutinib versus ibrutinib (78.3% versus 62.5%), especially in patients with del11q (83.6% versus 69.1%) and del(17p) (83.3% versus 53.8%), as were 12-mo PFS (94.9% versus 84.0%) and OS rates (97.0% versus 92.7%) [60]. Moreover, zanubrutinib has been studied in combination with venetoclax and Obinutuzumab (BOVen), showing deep responses with many patients attaining uMRD in a multicenter Phase I/II trial (NCT03824483) [61]. Concerning the safety of zanubrutinib, it has a safety profile superimposable to that of acalabrutinib, with neutropenia, upper respiratory tract infections, and pneumonia being the most frequent adverse events with relatively few serious events. The incidence of atrial fibrillation is as low as the one of acalabrutinib as well, while that of major hemorrhage is similar compared to ibrutinib. In distinction from acalabrutinib, zanubrutinib absorbance is not affected by taking proton-pump inhibitors (e.g., omeprazole, lansoprazole, pantoprazole, etc.), and it does not prolong QTc significantly. Pirtobrutinib is a highly selective, reversible BTKi, which blocks the ATP binding site of BTK with a noncovalent, non-C481-dependent binding. Recently, the BRUIN study—a phase 1/2 study of pirtobrutinib in mature B-cell malignancies—has shown promising results [62]. Of 323 patients, 170 with relapsed/refractory CLL were enrolled. The median age across the study was 68 years (range 62–74); 25% of patients harbored 17p deletion, 30% TP53 mutation, 19% 11q deletion, and 88% unmutated IGHV. The median number of previous lines of therapy was 3 (range 2–5), with 86% of patients already treated with a BTKi, 90% with an anti-CD20, 34% with venetoclax, 21% with PI3K inhibitor, 6% with CAR T-cell therapy, and 2% with allogeneic transplant. The treatment showed an ORR of 63% (95% CI 55–71) and, surprisingly, the efficacy of the drug not only showed no difference in the cytogenetic and molecular high-risk groups (ORR 79% in del(17p) and/or TP53mut patients) but showed also an equal ORR of 62% in BTKi-pretreated patients. Moreover, ORR was similar in patients who previously discontinued another BTKi for progression (67%) versus toxicity (52%), and in patients with a BTK C481 mutation (75%) and patients without (60%). These findings, put in the current treatment landscape, will offer a consistent option in the treatment sequencing of relapsed/refractory heavily pretreated patients with considerably low adverse events. The most common adverse events were fatigue (20%), diarrhea (17%), contusion (13%), and grade 3 or higher neutropenia (10%). To note, no cases of grade 3 atrial fibrillation/flutter were reported, while only one patient suffered major bleeding after mechanical trauma. The discontinuation rate due to adverse events was 1%. Although the current evidence is based on a population that is on average younger than the median age of diagnosis of CLL, we believe that these two drugs in particular will be able to fit into the therapeutic armamentarium for elderly/unfit patients due to the better toxicity profile demonstrated and the excellent response rate obtained, even in patients with unfavorable characteristics, compared with the registration trials of the drugs currently available to us.

**Table 2 jcm-10-05104-t002:** Currently available evidence from trials of selective inhibitors in elderly patients with CLL.

Treatment-Regimen	Study Type	Number of Patients	Setting	Median Age	Comorbidity	ORR	CR	PFS	Survival Benefit	Summary of Toxicities
**Ibrutinib**										
RESONATE-2 [38]	Phase 3	136	1°L	72	32% CIRS > 6	86%	4%	60-month PFS 89%	Yes	grade ≥ 3 neutropenia 10%, grade ≥ 3 infections NA, Diarrhea 42% (4% grade > 3)
RESONATE [39,40]	Phase 3	195	R/R	67	20% CIRS > 6	43%	0%	Median PFS 44.1 months	Yes	Grade ≥ 3 neutropenia 16%, grade ≥ 3 infections 24%, Bleeding 44% (1% grade > 3), Atrial fibrillation 3%
**Idelalisib–Rtx**										
O’Brien et al. [46]	Phase 2	64	1°L	71	NA	97%	19%	36-month PFS 83%	NA	grade ≥ 3 neutropenia 28%, grade ≥ 3 infections 9%, Diarrhea and/or colitis 42%, Pneumonia 19%,
CLL-R2 [47,48]	Phase 3	110	R/R	71	Median CIRS 8	81%	0%	Median PFS 20.3 months	Yes	grade ≥ 3 neutropenia 3%, grade ≥ 3 infections NA, Diarrhea
**Acalabrutinib**										
ELEVATE-TN [42]	Phase 3	179	1°L	70	12% CIRS > 6	86%	0%	30-month PFS 90%	No	Grade ≥ 3 neutropenia 9%, grade ≥ 3 infections 14%
ASCEND [43]	Phase 3	155	R/R	68	26% CrCl<60 mL/min, 13% ECOG≥2	81%	0%	12-month PFS 88%	Yes	grade ≥ 3 neutropenia 15%, grade ≥ 3 infections NA, Diarrhea 18% (1% grade > 3)
**Venetoclax**										
Stilgenbauer et al. (del(17p) patients) [50]	Phase 2	107	R/R	67	9% ECOG ≥ 2	85%	10%	24-month PFS 54%	NA	grade ≥ 3 neutropenia 40%, grade ≥ 3 infections 20%, AIHA 7%
**Venetoclax–Rtx**										
MURANO [7,51]	Phase 3	194	R/R	64	3% CrCl < 50 mL/min, 0.5% ECOG ≥ 2	93%	27%	48-month PFS 57.3%	Yes	Grade ≥ 3 neutropenia 58%, grade ≥ 3 infections 18%, Pneumonia 8%, TLS 3%
**G-Venetoclax**										
Fischer et al. [6]	Phase 3	216	1°L	72	86% CIRS > 6, 60% CrCl < 70 mL/min	85%	49%	24-month PFS 88.2%	Yes	grade ≥ 3 neutropenia 53%, grade ≥ 3 infections 18% IRR 9%
**Zanubrutinib**										
Xu et al. [58]	Phase 2	91	R/R	61	3.3% ECOG ≥ 2	84.6%	3.3%	12-month PFS 87.2%	NA	grade ≥ 3 neutropenia 44%, grade ≥ 3 thrombocytopenia 15.4%, pneumonia 13.2%, upper respiratory tract infection 9.9%, anemia 8.8%
SEQUOIA (Cohort C) [59]	Phase 3	109	1°L	70	12.8% ECOG ≥ 2	94.5%	3.7%	18-month PFS 88.6%	Yes	contusion 20.2%, upper respiratory tract infection 19.3%, neutropenia 17.4%, diarrhea 16.5%
ALPINE (interim analysis) [60]	Phase 3	415	R/R	age ≥ 65 62.3%	Not reported	78.3%	NA	12-month PFS 94.9%	Yes	Neutropenia 28.4%, grade ≥ 3 infection 12.7%, Atrial fibrillation 2.5%, Bleedings 2.9%
**Pirtobrutinib (LOXO-305)**										
BRUIN [62]	Phase 2	170	R/R	68	8% ECOG ≥ 2	63%	0%	12-month PFS 82%	NA	fatigue 20%, diarrhea 17%, contusion 13%, grade ≥ 3 neutropenia 10%

Abbreviations: 1°L, frontline treatment; AIHA, autoimmune hemolitic anemia; CIRS, Cumulative Illness Rating Scale; CR, complete response; CrCl, creatinine clearance; ECOG, Eastern Cooperative Oncology Group Performance Status; G-Venetoclax, obinutuzumab-venetoclax; Idelalisib-Rtx, idelalisib-riuximac; IRR, infusione-related reaction; NA, not available; ORR, overall response rate; PFS, progression-free survival; R/R, relapsed/refractory chronic lymphocytic leukemia; Venetoclax-Rtx, venetoclax-rituximab.

## 4. Appropriate Therapies for Older/Unfit Patients: Practical Guidance

### 4.1. Frontline Treatment

Treatment recommendations for patients treated outside clinical trials are summarized in Table 3. Elderly patients (aged 65 years or older) with good fitness status (i.e., CIRS <6, CrCl >70 mL/min, ECOG <2, no geriatric syndromes) and without prognostic adverse factors (i.e., mutated IGVH status, no TP53 mutation/deletion) may be considered for frontline therapy with BR. In fact, when compared with fludarabine-based regimens, BR showed a lower rate of infections and no difference in survival [25]. For unfit patients with mutated IgVH status and no TP53 mutation/deletion, there are several possible first-line treatments. Chlorambucil–obinutuzumab combination [5] and single-agent treatment with ibrutinib [63] are the safest and most effective options in this population, and also the most readily available. Of equal or even greater efficacy and less toxicity are the combination of venetoclax and obinutuzumab [6] and the second-generation BTK inhibitor acalabrutinib [42]; unfortunately, they are not yet approved or readily available in several states; however, they will most likely be used more and more in the near future due to their improved tolerability and fewer off-target side effects. In contrast with individuals with favorable cytogenetic risk, patients with unmutated IgHV and/or TP53 mutation/deletion should be treated first-line with agents that flatten to the point of nullifying the difference with patients that are at lower cytogenetic risk. Patients with unmutated IgHV should be preferably treated with BCR inhibitors, ibrutinib [39,40], and acalabrutinib [43], which show no differences in survival between mutated and unmutated IgHV patients. Patients harboring TP53 mutation/deletion should be treated with the BTK inhibitors ibrutinib or acalabrutinib (if available), or venetoclax if unsuitable for treatment with BCR inhibitors, alone or in combination with obinutuzumab (if approved and available).

### 4.2. Relapsed/Refractory

For patients with relapsed/refractory CLL, based on the current evidence, ibrutinib is the treatment of choice, regardless of cytogenetic risk [39]. If approved, considering the more favorable risk profile, acalabrutinib can be considered a valid alternative to ibrutinib, especially in these group of patients. Venetoclax is approved for relapsed/refractory CLL with TP53 mutation/deletion, and for CLL patients relapsing during treatment with BCR inhibitors, either alone [50] or preferably, if the patients’ fitness status permits, in combination with rituximab [7,51].

### 4.3. Treatment of the Elderly/Unfit Patients: An Overview

In the era of targeted therapies, treatment of elderly/unfit patients with CLL must be a balance between efficacy and safety. Although it is not possible to make direct comparisons between results of different studies, some considerations may guide the choice toward the best treatment option in elderly patients, considering both quality of life and life expectancy. Older patients are more susceptible to the myelotoxicity of chemoimmunotherapy than younger patients because of the age-related decline in hematopoietic reserve, so only fit patients without or with minimal geriatric syndromes will be eligible for candidacy. On the other hand, based on the evidence available to date, the treatment with inhibitors is preferred not only for unfit patients, but also for fit patients harboring unmutated IgVH and/or del(17p) and/or TP53 mutations/deletions. Whether a comprehensive geriatric assessment and the intervention of a multidisciplinary panel with a geriatrist and other needed specialists (neurologist, cardiologist, etc.) prior to or during therapy of CLL could support hematologists in the treatment choice improving tolerability or feasibility in elderly individuals is still an unanswered question [64], although we strongly suggest it. A correct framing of the possible “weak points” of patients in need of treatment can correctly direct the clinician towards a patient-tailored therapy avoiding predictable complications based on the toxicity profile of each of the available drugs. The improved tolerability in elderly/unfit patients of biological treatments, which are less myelotoxic, combined with their superior efficacy, even in patients with unfavorable molecular risk, will progressively improve their outcome in the near future. These considerations emphasize the importance of reconsidering the definition of fitness status to individualize treatment strategies. Hence, it is fundamental to consider the possible age-related adverse events, which could lead to discontinuation of the treatment, with the risk of loss of its efficacy. Several critical issues should be considered in the pretreatment evaluation, such as the indeterminate duration of most treatments, the specific toxicity profile (i.e., atrial fibrillation and bleeding with ibrutinib, autoimmune manifestations with idelalisib, IRR with obinutuzumab, autoimmune cytopenias, and TLS with venetoclax), patient compliance to treatment, and the goal of therapy (disease eradication versus clinical response). Second-generation drugs—including, for example, acalabrutinib, zanubrutinib, and pirtobrutinib—will help overcome some of these as the safety profile improves with greater formulation specificity. The achievement of a response, even if partial, that allows an effective control of the disease is sufficient to ensure an excellent quality of life in elderly and/or unfit patients. Continuous therapy may in fact be the best solution compared with fixed-term therapies both from the point of view of toxicity and of logistics (number of accesses to the hospital, need for close monitoring of blood tests, availability of caregivers, etc.), especially for these patients, although it remains to be verified in the future the real-life experience with the new chemo-free fixed-term therapies. Therefore, the next steps should be directed towards an increasingly tailored approach to these patients, through the development of trials that combine fitness status and treatment-specific toxicities, to ensure better profiling and improve the management of increasingly elderly and complex patients.

## Figures and Tables

**Table 1 jcm-10-05104-t001:** Currently available evidence from trials of chemoimmunotherapy in elderly patients with CLL.

Treatment- Regimen	Study Type	Number of Patients	Setting	Median Age	Comorbidity	ORR	CR	Median PFS	Survival Benefit	Summary of Toxicities
BR										
MABLE [26]	Phase 3	121	1°L	72	Median CIRS 3	91%	24%	40 months	No	grade ≥ 3 neutropenia 45%, grade ≥ 3 infections 19% (lower toxicity than FCR)
**G-Benda**										
GREEN [30]	Phase 3	158	1°L	69	18% CIRS > 6 and 46% CrCl < 70 mL/min	81%	35%	Not reached	NA	grade ≥ 3 neutropenia 53%, grade ≥3 infections 20%, IRR 17%
**G-Chl**										
CLL11 [5,31]	Phase 3	333	1°L	74	Median CIRS 8	79%	21%	29 months	Yes	grade ≥ 3 neutropenia 33%, grade ≥ 3 infections 12%, IRR 10%
**O-Chl**										
COMPLEMENT-1 [35]	Phase 3	221	1°L	69	Median CIRS 8	82%	14%	22 months	No	grade ≥ 3 neutropenia 26%, grade ≥ 3 infections 9%, IRR 10%

Abbreviations: 1°L, frontline treatment; BR, bendamustine-rituximab; CIRS, Cumulative Illness Rating Scale; CR, complete response; CrCl, creatinine clearance; FCR, fludarabine-cyclophosphamide-rituximab; G-Benda, obinutuzumab-bendamustine; G-Chl, obinutuzumab-chlorambucil; IRR, infusione-related reaction; O-Chl, ofatumumab-chlorambucil; ORR, overall response rate; PFS, progression-free survival.

**Table 3 jcm-10-05104-t003:** Current treatment recommendations in elderly (older than 65 years) patients with CLL.

	FIT Patients (CIRS < 6 and CrCl > 70 mL/min)	UNFIT Patients (CIRS > 6 and or CrCl < 70 mL/min)
**FRONTLINE TREATMENT**		
**IGHV mutated and ** **TP53 mutation/deletion absent**	**Bendamustine**–**Rituximab** **(standard dose or dose-modified)**	**Ibrutinib**
	**Ibrutinib**	**Chlorambucil**–**Obinutuzumab**
		Venetoclax–Obinutuzumab (if approved and available)
		Acalabrutinib (if available)
**IGHVunmutated and ** **TP53 mutation/deletion absent**	**Ibrutinib**	**Ibrutinib**
	Bendamustine–Rituximab (standard dose or dose-modified)	Venetoclax–Obinutuzumab (if approved and available)
		Acalabrutinib (if available)
		Chlorambucil–Obinutuzumab
**TP53 mutation/deletion present**	**Ibrutinib**	
	**Venetoclax monotherapy**	
	Venetoclax–Obinutuzumab (if approved and available)	
	Acalabrutinib (if available)	
	Idelalisib–Rituximab	
**RELAPSED/REFRACTORY**		
**TP53 mutation/deletion absent**	**Ibrutinib**	
	**Venetoclax**–**Rituximab**	
	**Venetoclax monotherapy**	
	Acalabrutinib (if available)	
	Idelalisib–Rituximab	
**TP53 mutation/deletion present**	**Ibrutinib**	
	**Venetoclax**–**Rituximab**	
	Acalabrutinib (if available)	
	Venetoclax monotherapy	
	Idelalisib–Rituximab	

Abbreviations: CIRS, Cumulative Illness Rating Scale; CrCl, creatinine clearance; IGHV, immunoglobulin heavy chain variable.

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
