# Peer review of "Treatment Options for Elderly/Unfit Patients with Chronic Lymphocytic Leukemia in the Era of Targeted Drugs: A Comprehensive Review"

_jcm, 2021, doi:10.3390/jcm10215104_

Round 1

Reviewer 1 Report

General remarks:

I miss the definitions of terms used in the paper like Minimal residual disease (MRD). The parameters like Cumulative Illness Rating Scale (CIRS) and CrCl should be introduced at the beginning and defined. Other examples of such practice is described in specific comments below.

Besides I suggest that the authors introduce and define the patient' fitness status. It is defined in table 3 line 391. It would be easier to follow the paper it the definition is given at the beginning of the paper.

In my opinion too mamy abbreviations make the reading of the paper very difficult for the reader that is not from the field. Without the explanations of the abbreviations and short definitions of the meaning of the terms limit the  reading audience of the review and in my opinion, decreases citing potential of the presented review paper.

The acronyms of the clinical trials should be specified and the references on clinical trials should be added.

The general organization of the paper should be improved.

Specific comments are:

1. Introduction

The definition of Minimal residual disease (MRD) negativity is missing.

The statement: “MRD negativity is mainly achieved with time-limited therapies, while continuous therapies show a high response rate, although mainly partial, and a significant increase in disease-free survival.” Is not clear.

What about continuous therapies: “They have a high response rate, although mainly partial, and a significant increase in disease-free survival.” It seems contradictory. The clarification is needed.  If the continuous therapy is the effective control of disease it would be useful to know how this efficiency is measured?

I would suggest the authors to pay more attention to the structure of the introduction and to present the information in the introduction in the organized way: a) charateristics of the disease, b) the differences between younger and elredy patients c) ways of treatments/types of therapies c) expected outcomes d) the aim and the impact of the presented review.

The explanation of the context as stated now is hard to understand.

The introduction should contain more references to help the reader find the source of written statements.

2. The evaluation of patients’ fitness status

The section should be rewritten and organized in the logic manner. I would first introduce different scales of assessment than compare them! When first mentioned in the text abbreviations should be introduced with the full name and defined. Specific cases of unclarity are for example:

The meaning of CIRS, ECOG and the scales. Brief definition is needed so the reader can follow the text.

Similar situation occurs with BR. Please define the abbreviations when first used in the text, so the reader do not need to guess the meaning.

For clinical trials please explain the tials in brief and add references for all the trails discussed in the paper.

What is PFS and EFS? Explain the CIRS levels. What does it mean CIRS > 6 in the sentence ”CIRS>6 (the definition should be given at the beginning of the paper)

It is not clear if the profiling of patients or their fitness status are important! The sentence is: “a better profiling of patients’ fitness status……”  does it means that one should profile the fitness status or to choose more fit patients?

CGA Please explain the abbreviation and add short definition of CGA?

3. Therapeutic approaches in older and/or unfit patient population

It is not clear if the same approaches are suitable for older and unfit patients? Or for older or unfit patients. What is the age limit for unfit younger patients to follow the same criteria as elderly patients?  Please use the same term for the same category: geriatric population, old, elderly.. if the terms are not exchangeable please determine/define the differences.

3.1 Chemoimunotherapy

I miss the definition of the fit patient. How do you discriminate the fit and unfit patient? Is the CIRS score 2 the discrimination measure?

The definition of the progression-free survival (PFS) is missing. I suggest the definition of progression measure at the beginning of the paper so the reader can follow the logic of the review.

FCR

Too many abbreviations and a lot of information in one sentence make it almost impossible to follow the logic of the written facts. Example from the text: “Of note, retrospective multicenter experiences in elderly and/or unfit CLL patients in Italy before the approval of Chlorambucil-Obinutuzumab (G-Chl) regimen have highlighted and evaluated the need for dose adjustment of the previously used chemoimmunotherapy regimens [19-21], showing that patients obtaining a CR could benefit from lighter regimens (e.g., by reducing the dose per cycle or the number of cycles), but otherwise RDI reduction significantly affects both PFS and OS [9].”

Is the GREEN acronym for the try or a special sort of trail? Please explain and add reference to the clinical trial!

Table 1 should contain the references to the presented data. I suggest the new column with references on the clinical trial.

The explanation and a brief definition of the minimum anticipated biological effect level (MABEL) is required. Without the explanations the audience of the review is limited to the experts in the field and in my opinion decreases citing  potential of the presented review paper.

3.2 BCR inhibitors

Please explain the abbreviations and before explaining the specific BCR inhibitors please provide some common information on them.

BTK explain the abbreviation and provide some data on BTK.   

Explain/define that a non-inferiority trials provide a direct comparison of the relative benefit-to-risk profiles of a new vs standard treatment.

Define the large retrospective multicenter studies [47,48] their aim, the geographic zone (USA and UK?) the benefits of such studies… differences with conventional clinical trials…

Explain the abbreviation TLS

The definition of older and younger patients would enable the reader to understand in which age ranges the authors are talking about efficacy and safety.

The sentence 273-276 “To compare efficacy and safety in older and younger patients, a large multicentric retrospective study on 342 patients comparing age categories ≥75 and <75 years treated in the elapsed/refractory non-trial setting has been conducted51”  The same sentence will in my opinion be easier to read if the authors straight ahead define the younger patients  ≥75 and the older patients  over <75 years.  A number 51 at the end of the sentence is probably the reference to the cited research.

Define ORR, PFS. Maybe it was defined already but too many abbreviations  is the sentence: “No differences in terms of ORR (81% vs 82%), 1-year PFS (79% vs 73%) and 1-year OS (77% vs 83%) have been observed, not even by performing an analysis adjusted for mono versus combination venetoclax-based therapy.” Make it realy difficult to understand the message. In my opinion the abbreviation should be kept at minimum and standardized so the reading can be fluent. As written now the reader have to have a table with the abbreviations to follow the writings.

3.3 and 3.4

Lines 265-266 “In the real-life setting, large retrospective multicentre 265 studies [47,48] have demonstrated reassuringly similar efficacy and survival to trial out-266 comes.” The sentence is not clear. I suggest the authors add the clinical trail number for all the trials like they do in line 332-333:

“multi-center Phase I/II 332 trial (NCT03824483) [56]”

Define and explain CrCl.

The definitions of fit and unfit patients are given in line 391 in Table 3 should be given earlier in the paper.

Author Response

Dear Reviewer,

Thank you for your kind suggestions and the opportunity to revise our paper “Treatment Options for Elderly/Unfit Patients with Chronic Lymphocytic Leukaemia in the Era of Targeted Drugs: a com-prehensive review”. The suggestions offered have been immensely helpful.

I have included the comments immediately after this letter and responded to them individually, indicating exactly how we addressed each concern or problem and describing the changes we have made. The revisions have been approved by all authors. The changes are marked in red in the paper as requested.

We hope the revised manuscript will better suit the Journal of Clinical Medicine but are happy to consider further revisions, and we thank you for your continued interest in our research.

Sincerely,

Alberto Fresa, MD

Sezione di Ematologia, Dipartimento di Scienze Radiologiche ed Ematologiche

Università Cattolica del Sacro Cuore

Point 1: Introduction

The definition of Minimal residual disease (MRD) negativity is missing. The statement: “MRD negativity is mainly achieved with time-limited therapies, while continuous therapies show a high response rate, although mainly partial, and a significant increase in disease-free survival.” Is not clear. What about continuous therapies: “They have a high response rate, although mainly partial, and a significant increase in disease-free survival.” It seems contradictory. The clarification is needed.  If the continuous therapy is the effective control of disease it would be useful to know how this efficiency is measured? I would suggest the authors to pay more attention to the structure of the introduction and to present the information in the introduction in the organized way: a) charateristics of the disease, b) the differences between younger and elredy patients c) ways of treatments/types of therapies c) expected outcomes d) the aim and the impact of the presented review. The explanation of the context as stated now is hard to understand. The introduction should contain more references to help the reader find the source of written statements.

Response 1: Thank you for your suggestions. We have included an overview of fitness status assessment and first-line therapy in the introduction, although we decided to refer the specifics of each treatment to the text to avoid making it too verbose and repetitive. We have also included the definition of MRD, better contextualizing the goals of the review.

Point 2: The evaluation of patients’ fitness status

The section should be rewritten and organized in the logic manner. I would first introduce different scales of assessment than compare them! When first mentioned in the text abbreviations should be introduced with the full name and defined. Specific cases of unclarity are for example: The meaning of CIRS, ECOG and the scales. Brief definition is needed so the reader can follow the text. Similar situation occurs with BR. Please define the abbreviations when first used in the text, so the reader do not need to guess the meaning. For clinical trials please explain the tials in brief and add references for all the trails discussed in the paper. What is PFS and EFS? Explain the CIRS levels. What does it mean CIRS > 6 in the sentence ”CIRS>6 (the definition should be given at the beginning of the paper) It is not clear if the profiling of patients or their fitness status are important! The sentence is: “a better profiling of patients’ fitness status……”  does it means that one should profile the fitness status or to choose more fit patients? CGA Please explain the abbreviation and add short definition of CGA?

Response 2: Thank you for your precious suggestions. We decided to add a brief introduction to the section 2 to improve the understanding of the successive evaluations, giving definitions and descriptions of the scales. Moreover we explained all the abbreviations. Finally, concerning the sentence “a better profiling of patients’ fitness status……” , we edited the sentence to allow the reader to better understand that we intended to profile the fitness status to adapt therapy choices.

Point 3. Therapeutic approaches in older and/or unfit patient population

It is not clear if the same approaches are suitable for older and unfit patients? Or for older or unfit patients. What is the age limit for unfit younger patients to follow the same criteria as elderly patients?  Please use the same term for the same category: geriatric population, old, elderly.. if the terms are not exchangeable please determine/define the differences.

Response 3: As suggested before, we gave the definitions at the beginning of the manuscript, and we explicated them also at the beginning of this section.

Point 4. Chemoimunotherapy

I miss the definition of the fit patient. How do you discriminate the fit and unfit patient? Is the CIRS score 2 the discrimination measure? The definition of the progression-free survival (PFS) is missing. I suggest the definition of progression measure at the beginning of the paper so the reader can follow the logic of the review. Too many abbreviations and a lot of information in one sentence make it almost impossible to follow the logic of the written facts. Example from the text: “Of note, retrospective multicenter experiences in elderly and/or unfit CLL patients in Italy before the approval of Chlorambucil-Obinutuzumab (G-Chl) regimen have highlighted and evaluated the need for dose adjustment of the previously used chemoimmunotherapy regimens [19-21], showing that patients obtaining a CR could benefit from lighter regimens (e.g., by reducing the dose per cycle or the number of cycles), but otherwise RDI reduction significantly affects both PFS and OS [9].” Is the GREEN acronym for the try or a special sort of trail? Please explain and add reference to the clinical trial! Table 1 should contain the references to the presented data. I suggest the new column with references on the clinical trial. The explanation and a brief definition of the minimum anticipated biological effect level (MABEL) is required. Without the explanations the audience of the review is limited to the experts in the field and in my opinion decreases citing  potential of the presented review paper.

Response 4: As suggested before, we gave the definitions at the beginning of the manuscript, and we explicated them also at the beginning of this section. We reduced the abbreviations as much as possible, and split the long sentences in two or three, to make the readers follow the logic of the facts. Moreover, we added the references next to the names of the trials, both in the text and in the table. Concerning MABEL, we think there must have been a misunderstanding, as we cited the MABLE trial, comparing Benda-R to Chl-R.

Point 5. BCR inhibitors

Please explain the abbreviations and before explaining the specific BCR inhibitors please provide some common information on them. BTK explain the abbreviation and provide some data on BTK.  

Explain/define that a non-inferiority trials provide a direct comparison of the relative benefit-to-risk profiles of a new vs standard treatment. Define the large retrospective multicenter studies [47,48] their aim, the geographic zone (USA and UK?) the benefits of such studies… differences with conventional clinical trials… Lines 265-266 “In the real-life setting, large retrospective multicentre studies [47,48] have demonstrated reassuringly similar efficacy and survival to trial outcomes.” The sentence is not clear. I suggest the authors add the clinical trail number for all the trials like they do in line 332-333:“multi-center Phase I/II 332 trial (NCT03824483) [56]”

Explain the abbreviation TLS. The definition of older and younger patients would enable the reader to understand in which age ranges the authors are talking about efficacy and safety. Define and explain CrCl. The definitions of fit and unfit patients are given in line 391 in Table 3 should be given earlier in the paper.

The sentence 273-276 “To compare efficacy and safety in older and younger patients, a large multicentric retrospective study on 342 patients comparing age categories ≥75 and <75 years treated in the elapsed/refractory non-trial setting has been conducted51”  The same sentence will in my opinion be easier to read if the authors straight ahead define the younger patients  ≥75 and the older patients  over <75 years.  A number 51 at the end of the sentence is probably the reference to the cited research. Define ORR, PFS. Maybe it was defined already but too many abbreviations  is the sentence: “No differences in terms of ORR (81% vs 82%), 1-year PFS (79% vs 73%) and 1-year OS (77% vs 83%) have been observed, not even by performing an analysis adjusted for mono versus combination venetoclax-based therapy.” Make it realy difficult to understand the message. In my opinion the abbreviation should be kept at minimum and standardized so the reading can be fluent. As written now the reader have to have a table with the abbreviations to follow the writings.

Response 5: All the abbreviations (BCR, BTK, CrCl, ORR, PFS, TLS) have been explicated throughout the section. We added the definition of non-inferiority trial and provided some information about the retrospective studies, explaining the key findings especially regarding safety, a crucial issue to address in comparison to clinical trials. Concerning the distinction between older and younger patients, throughout the manuscript we have considered as “old” patients older than 65 years, and as suggested, we provided this definition at the beginning of the manuscript. In the sentence regarding the results of study #55, the authors increased the baseline age to show any differences on patients older than 75 years, which is why we have made the baseline age explicit in this passage.

Reviewer 2 Report

The manuscript entitled ‘Treatment Options for Elderly/Unfit Patients with Chronic Lymphocytic Leukaemia in the Era of Targeted Drugs: a comprehensive reviewis not suitable to be published in “Journal of Clinical Medicine”. It is understood from this Review that the authors discussed treatment decisions in elderly and/or unfit patients with chronic lymphocytic leukemia (CLL). Although the idea is good, this Review does not reflect the main goal influentially. Sections are written so independently from each other and repeats themselves.

Here are the some points to be corrected:

  1. “Regarding the gender distribution significantly more men develop CLL in comparison to women (1.9-fold higher incidence), which remains stable also in higher age groups”. This sentence must be changed as “Regarding the gender distribution significantly men develop CLL more frequently in comparison to women (1.9-fold higher incidence), which remains stable also in higher age groups”.
  2. “the development of new non-chemotherapy drugs with a safer toxicity profile and proven efficacy” must be changed as “the development of new non-chemotherapy drugs with safer toxicity profiles and proven efficacy”.
  3. In section 2.1., the meaning of CIRS, ECOG must be given and the connection of them with CLL must be explained well.
  4. In section 2.2, “The” must be given with a lower case in line 90.
  5. “A score of more rapid use, evaluated on a population of patients with CLL, was shown by Molica S. et al., who combining ADL, IADL and CIRS showed that patients with worse performance have shorter survival” this sentence must be rewritten.
  6. “Rituximab (BR)” is given in section 3.1. However, it was mentioned in earlier sections so it is complicated to understand for readers.
  7. “The meaning of FCR, RDI and OS” must be given in section 3.1.
  8. In “(e.g. bendamustine dosed with 70 mg/m2 instead of 90 mg/m2)”, m2 must be superscript.
  9. Both Table 1 and Table 2 must be corrected and distributed rows and columns evenly.
  10. In section 3.2.1. “a first-in-class inhibitor of Bruton’s tyrosine kinase”, there must be added the abbreviation of “Bruton’s tyrosine kinase (BTK)”. Because, BTK was mentioned in further sections so often.
  11. “though CYP3A4 inhibitors use” in line 194, these inhibitors must be specified.
  12. The generic names of drugs must be written with a lower case at least there must be a standard. The upper case and lower case were both used in the manuscript.
  13. “incidencc” must be corrected as “incidence”.
  14. “deatsh” must be corrected as “deaths”.
  15. “conducted51” must be corrected as “conducted [51]”.
  16. “proton-pump inhibitors” in line 329 must be specified.
  17. “The content of the sections 3,4,5” were quiet similar, they must be combined.
  18. “n eng j med” must be corrected as “N Eng J Med” in 499.
  19. The Reference 53 was written totally wrong with the name, volume and issue. The journal name was forgetten. The correct form is “Xu W, Yang S, Zhou K, et al. Treatment of relapsed/refractory chronic lymphocytic leukemia/small lymphocytic lymphoma with the BTK inhibitor zanubrutinib: phase 2, single-arm, multicenter study. J Hematol Oncol. 13(1):48. 

Author Response

Dear Reviewer,

Thank you for your kind suggestions and the opportunity to revise our paper “Treatment Options for Elderly/Unfit Patients with Chronic Lymphocytic Leukaemia in the Era of Targeted Drugs: a com-prehensive review”. The suggestions offered have been immensely helpful.

I have included the comments immediately after this letter and responded to them individually, indicating exactly how we addressed each concern or problem and describing the changes we have made. The revisions have been approved by all authors. The changes are marked in red in the paper as requested.

We hope the revised manuscript will better suit the Journal of Clinical Medicine but are happy to consider further revisions, and we thank you for your continued interest in our research.

Sincerely,

Alberto Fresa, MD

Sezione di Ematologia, Dipartimento di Scienze Radiologiche ed Ematologiche

Università Cattolica del Sacro Cuore

Point 1: Introduction

Regarding the gender distribution significantly more men develop CLL in comparison to women (1.9-fold higher incidence), which remains stable also in higher age groups”. This sentence must be changed as “Regarding the gender distribution significantly men develop CLL more frequently in comparison to women (1.9-fold higher incidence), which remains stable also in higher age groups”; “the development of new non-chemotherapy drugs with a safer toxicity profile and proven efficacy” must be changed as “the development of new non-chemotherapy drugs with safer toxicity profiles and proven efficacy”.

Response 1: Thank you for your suggestions. We have edited the sentences as suggested.

Point 2: The evaluation of patients’ fitness status

In section 2.1., the meaning of CIRS, ECOG must be given and the connection of them with CLL must be explained well. In section 2.2, “The” must be given with a lower case in line 90. “A score of more rapid use, evaluated on a population of patients with CLL, was shown by Molica S. et al., who combining ADL, IADL and CIRS showed that patients with worse performance have shorter survival” this sentence must be rewritten.

Response 2: Thank you for your suggestions. We decided to add a brief introduction to the section 2 to improve the understanding of the successive evaluations, giving definitions and descriptions of the scales. We corrected the typo and edited the sentences as requested.

Point 3: Therapeutic approaches

“Rituximab (BR)” is given in section 3.1. However, it was mentioned in earlier sections so it is complicated to understand for readers. “The meaning of FCR, RDI and OS” must be given in section 3.1. In section 3.2.1. “a first-in-class inhibitor of Bruton’s tyrosine kinase”, there must be added the abbreviation of “Bruton’s tyrosine kinase (BTK)”. Because, BTK was mentioned in further sections so often.

Response 3: We have edited the section 3 according to the suggestions.

Point 4: In “(e.g. bendamustine dosed with 70 mg/m2 instead of 90 mg/m2)”, m2 must be superscript. The generic names of drugs must be written with a lower case at least there must be a standard; the upper case and lower case were both used in the manuscript. “incidencc” must be corrected as “incidence”; “deatsh” must be corrected as “deaths”; “conducted51” must be corrected as “conducted [51]”. “n eng j med” must be corrected as “N Eng J Med” in 499.

Response 4: All the errors reported have been corrected.

Point 5: Both Table 1 and Table 2 must be corrected and distributed rows and columns evenly.

Response 5: We edited the tables to better fit the journal layout.

Point 6: “though CYP3A4 inhibitors use” in line 194, these inhibitors must be specified.

Response 6: We have specified some inhibitors

Point 7: “proton-pump inhibitors” in line 329 must be specified.

Response 7: We have specified some inhibitors

Point 8: “The content of the sections 3,4,5” were quiet similar, they must be combined.

Response 8: Thank you for the precious advice! We have combined sections 4 and 5 to give a complete overview of the treatment. However, we considered it more effective to maintain separate the discussion of each treatment approach.

Point 9: The Reference 53 was written totally wrong with the name, volume and issue. The journal name was forgetten. The correct form is “Xu W, Yang S, Zhou K, et al. Treatment of relapsed/refractory chronic lymphocytic leukemia/small lymphocytic lymphoma with the BTK inhibitor zanubrutinib: phase 2, single-arm, multicenter study. J Hematol Oncol. 13(1):48.

Response 9: We corrected the reference.

Round 2

Reviewer 2 Report

The revised manuscript entitled ‘Treatment Options for Elderly/Unfit Patients with Chronic Lymphocytic Leukaemia in the Era of Targeted Drugs: a comprehensive reviewis suitable to be published in “Journal of Clinical Medicine”. Authors made corrections point by point and expanded the data properly in the manuscript.